# LEARNING DYNAMICS ON MANIFOLDS WITH NEURAL ORDINARY DIFFERENTIAL EQUATIONS

## ABSTRACT

Neural ordinary differential equations (Neural ODEs) have garnered significant attention for their ability to efficiently learn dynamics from data. However, for high-dimensional systems, capturing dynamics remains to be a challenging task. Existing methods often rely on learning ODEs on low-dimensional manifolds but usually require the knowledge of the manifold. Nevertheless, such knowledge is usually unknown in many scenarios. Therefore, we propose a novel approach to jointly learn data dynamics and the underlying manifold. Specifically, we employ an encoder to project the original data into the manifold and leverage the Jacobian matrix of its corresponding decoder for recovery. Our experimental evaluations encompass multiple datasets, where we compare the accuracy, number of function evaluations (NFE), and convergence speed of our model against existing baselines. Our results demonstrate superior performance, underscoring the effectiveness of our approach in addressing the challenges of high-dimensional dynamic learning.

## 1 INTRODUCTION

Understanding and modeling the dynamics of complex systems is a fundamental challenge in various fields, from physics to biology and engineering. To learn the dynamics, there are two basic components that need to be considered. The first is to learn a latent representation of the state of the system, and another is to learn how the latent state representation evolves forward in time (Floryan & Graham, 2022).

Neural ordinary differential equations (Neural ODEs) (Chen et al., 2018) have emerged as a powerful framework for learning dynamics from data efficiently. Their continuous-time modeling capabilities make them particularly suited for interpreting how the latent state representation evolves over time. The core idea of Neural ODEs is to use a neural network to parameterize a vector field, which is typically represented by a simple neural network (Haber & Ruthotto, 2017; Chen et al., 2018; Kidger, 2022). The neural network considers the current state of the system as input and produces the time derivative of that state as output, which determines how the system will change over time. By integrating the vector field over time, it is possible to calculate the system's trajectory and make predictions about its future behavior.

However, when dealing with high-dimensional spaces and complex, unknown dynamics, capturing accurate representations remains a challenging task. Existing methods often resort to numerical integration techniques (Pal et al., 2021; Daulbaev et al., 2020; Liu et al., 2021) or operate in higher-dimensional spaces (Dupont et al., 2019), which can either lead to increased computational complexity or introduce bias into the modeling process. Recent work (Lou et al., 2020) proposes to implicitly parameterize the original space with fewer parameters in the manifold. However, this requires the knowledge of the manifold.

One promising avenue to address these challenges lies in the field of manifold learning (Lou et al., 2020; Floryan & Graham, 2022; Lin & Zha, 2008), a powerful approach that enables us to capture and represent the underlying structure of high-dimensional data. A key assumption sometimes called the manifold hypothesis (Fefferman et al., 2016), is that the data lie on or near a low-dimensional manifold in state space. Manifold learning techniques aim to uncover the intrinsic low-dimensional manifolds within complex, high-dimensional datasets. By doing so, they provide valuable insights into the underlying dynamics of systems.

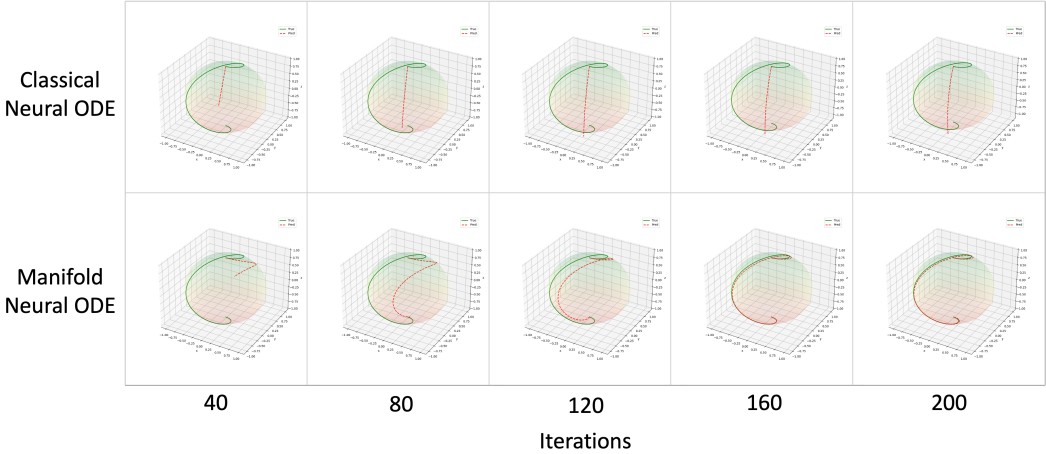

Figure 1: The top row of diagrams represents the dynamic learning process of classical Neural ODEs. The bottom row of diagrams represents the dynamics learning process on manifolds with Neural ODEs. The solid line represents the dynamics to be learned. The dotted line represents the dynamics that leaned by classical Neural ODEs and Manifold Neural ODEs with the increase of iterations.

In this work, we propose an innovative data-driven approach tailored to learning dynamics. By harnessing the principles of manifold learning, we focus on how the dynamics evolve in the most representative manifold space. This not only reduces the complexity of dynamic learning but also ensures the preservation of accuracy, a critical aspect when dealing with real-world data. Our method leverages a spatial encoder to obtain the latent state representation in manifold space from the original space. The Jacobian matrix of the encoder is also obtained and used for mapping back. A Neural ODE learns how the latent state evolves in the manifold space over time. We derive that the inverse of the Jacobian matrix can be used to map from the manifold space to the original space. In practice, the inverse of the Jacobian matrix can be replaced by a Jacobian matrix of the corresponding decoder of the encoder. A visual example is shown in Figure 1.

Our approach offers a unique solution to the challenges of dynamic learning in high-dimensional spaces by reducing data to the intrinsic dimensionality of the nonlinear manifold they live on. This is achieved by combining the rigorous mathematical theory of ordinary different equations in manifolds with the universal approximation capability of neural networks.

## 2 RELATED WORK

**Neural ODEs.** The basic idea of neural ordinary differential equations was originally considered in Rico-Martinez et al. (1992); Rico-Martinez & Kevrekidis (1993); Rico-Martinez et al. (1994). After Chen et al. (2018) specified the architecture of Neural ODEs and led to an explosion of applications in dynamic learning. For example, image classification (Dupont et al., 2019; Zhu et al., 2021), time series prediction (Jia & Benson, 2019; Norcliffe et al., 2020; Morrill et al., 2021; Guo et al., 2023), time series classification (Kidger et al., 2020), and continuous normalizing flows (Du et al., 2022). According to Chen et al. (2018), the scalar-valued loss with respect to all inputs of any ODE solver can be computed directly without backpropagating through the operations of the solver. The intermediate quantities of the forward pass will not need to be stored. It causes the Neural ODEs can be trained with a constant memory cost.

It is worth noting that the Neural ODEs are not universal approximators as shown by Dupont et al. (2019); Zhang et al. (2020). They address this limitation by introducing Augmented Neural ODEs, which add extra dimensions to vector field learning. This approach increases the degrees of freedom of the trajectory by elevating the dimensionality, thus solving the problem that trajectories cannot be crossed. However, it also introduces bias in learning the dynamics due to the additional dimensions.

**Efficiency of Neural ODEs.** As a continuous infinite-depth architecture, Neural ODEs will bring several drawbacks. The obvious drawback is that Neural ODEs have a low training efficiency (Dupont et al., 2019; Norcliffe et al., 2020; Pal et al., 2021; Daulbaev et al., 2020; Finlay et al., 2020; Lehtimäki et al., 2022; Djeumou et al., 2022). To accelerate the training speed, several works have been done. Some works try to improve the efficiency of ODE solvers, such as regularizing the solver (Pal et al., 2021), using interpolation backward dynamic methods (Daulbaev et al., 2020), or using second order ODE optimizer (Liu et al., 2021). Some works aim to optimize the objective function (Kelly et al., 2020; Ghosh et al., 2020; Pal et al., 2021; Xia et al., 2021). Simpler dynamics can lead to faster convergence and fewer discretization of the solver (Finlay et al., 2020).

One way to optimize the objective function is to take approximations of the learned dynamics. For example, (Finlay et al., 2020) demonstrated that appropriate regularization of the learned dynamics can significantly accelerate training time without degrading performance. However, these approaches may be less accurate when encountering unsmooth dynamics, such as those with more oscillations or abrupt changes. Other works are dedicated to optimizing the model structures, such as compressing the model. For example, Lehtimäki et al. (2022) used model order reduction to obtain a smaller-size NODE model with fewer parameters. However, optimizing the model structure by compressing the model without considering the characteristics of the data can result in a model with poor generalization capabilities.

**Neural ODEs on Manifolds.** Manifold learning is a subfield of machine learning and dimensionality reduction that focuses on discovering the underlying structure or geometry of high-dimensional data. The central idea behind manifold learning is that many real-world datasets lie on or near lower-dimensional manifolds within the high-dimensional space (Floryan & Graham, 2022). Lin & Zha (2008) formulate the dimensionality reduction problem as a classical problem in Riemannian geometry. For dynamic learning, Hairer (2011) describes the differential equation on the manifold. Its solution evolves on a manifold, and the vector field is often only defined on this manifold. Floryan & Graham (2022) explores the dynamics learning in the manifold using auto-encoder. Our work utilizes the Neural ODEs to learn better continuous dynamics.

Other works (Lou et al., 2020; Falorsi & Forré, 2020; Rozen et al., 2021; Gemici et al., 2016; Mathieu & Nickel, 2020) also investigate manifold generalization of Neural ODEs. These works calculate either the change in probability with a Riemannian change of variables, or the change through the use of charts and Euclidean change of variables. However, they are designed for normalizing flows, but the classification or regression task still remains to be investigated.

## 3 PRELIMINARY

### 3.1 MANIFOLDS

**Topological manifolds.** A topological space $\mathcal{M}$ is a topological manifold of dimension $d$ if it satisfies the following conditions: It is a second-countable Hausdorff space, ensuring that points can be separated by neighborhoods and that the topological structure is not too large. It is locally Euclidean of dimension $d$, meaning that at every point on the manifold, there exists a small neighborhood where the space behaves like Euclidean space. Furthermore, Whitney's embedding theorem (Whitney, 1936) states that any $d$-dimensional manifold $\mathcal{M}^d$ can be embedded in $\mathbb{R}^{2d+1}$. This means that a space of at most $2d + 1$ dimensions is sufficient to represent a $d$-dimensional manifold.

**Differentiable manifolds.** A topological manifold $\mathcal{M}$ is referred to as a smooth or differentiable manifold if it has the property of being continuously differentiable to any order. This implies that smooth functions can be defined on the manifold, making it suitable for calculus operations.

**Definition 3.1** (Smooth mapping). *Consider two open sets, $U \subset \mathbb{R}^r$ and $V \subset \mathbb{R}^s$, and let $\mathcal{G} : U \to V$ be a function such that for $x \in U$ and $y \in V$, $\mathcal{G}(x) = y$. If the function $\mathcal{G}$ has finite first-order partial derivatives, $\frac{\partial y_j}{\partial x_i}$, for all $i = 1, 2, \cdots, r$, and all $j = 1, 2, \ldots, s$, then $\mathcal{G}$ is said to be a smooth (or differentiable) mapping on $U$. We also say that $\mathcal{G}$ is a $\mathcal{C}^1$-function on $U$ if all the first-order partial derivatives are continuous. More generally, if $\mathcal{G}$ has continuous higher-order partial derivatives, $\frac{\partial^{k_1 + \cdots + k_r} y_j}{\partial x_1^{k_1} \cdots \partial x_r^{k_r}}$, for all $j = 1, 2, \cdots, s$ and all non-negative integers $k_1, k_2, \cdots, k_r$ such that $k_1 + k_2 + \cdots + k_r \leq r$, then we say that $\mathcal{G}$ is a $\mathcal{C}^r$-function, where $r = 1, 2, \cdots$.*

**Definition 3.2** (Diffeomorphism). *If $\mathcal{G}$ is a homeomorphism from an open set $U$ to an open set $V$, then $\mathcal{G}$ is said to be a $C^r$ diffeomorphism if both $\mathcal{G}$ and its inverse $\mathcal{G}^{-1}$ are $C^r$-functions.*

**Definition 3.3** (Diffeomorphic). *We say that $U$ and $V$ are diffeomorphic if there exists a diffeomorphism between them.*

Following the Definition 3.1, Definition 3.2, and Definition 3.3, we can straightforwardly extend these concepts to define diffeomorphism and diffeomorphic in manifolds (Ma & Fu, 2011).

**Definition 3.4** (Diffeomorphism in manifolds). *If $X$ and $Y$ are both smooth manifolds, a function $\mathcal{G} : X \to Y$ is a diffeomorphism if it is a homeomorphism from $X$ to $Y$ and both $\mathcal{G}$ and $\mathcal{G}^{-1}$ are smooth.*

**Definition 3.5** (Diffeomorphic of manifolds). *Smooth manifolds $X$ and $Y$ are diffeomorphic if there exists a diffeomorphism between them. In this case, $X$ and $Y$ are essentially indistinguishable from each other.*

## 3.2 NEURAL ODES

Neural ODEs are a family of deep neural network models that can be interpreted as a continuous version of Residual Networks (He et al., 2016). Recall the formulation of a residual network:

$$h_{t+1} - h_t = f(h_t, \theta_f), \tag{1}$$

where the $f$ is the residual block and the $\theta_f$ represents the parameters of $f$. The left side of Equation 1 can be seen as a denominator of 1, so it can be represented by $\frac{h_{t+1}-h_t}{1} = f(h_t, \theta_f)$. When the number of layers becomes infinitely large and the step becomes infinitely small, Equation 1 will become an ordinary differential equation format as shown in Equation 2.

$$\lim_{dt \to 0} \frac{h_{t+dt} - h_t}{dt} = \frac{dh(t)}{dt} = f(h(t), t, \theta_f). \tag{2}$$

Thus, the Neural ODE will have the same format as an ODE: $h'(t) = f(h(t), t, \theta_f)$ and $h(0) = x_0$, where $x_0$ is the input data. Typically, $f$ will be some standard simple neural architecture, such as an MLP. The $\theta_f$ represents trainable parameters in $f$. To obtain any final state of $h(t)$ when $t = T$, all that is needed is to solve an ordinary differential equation with initial values, which is called an initial value problem (IVP):

$$h(T) = h(0) + \int_0^T f(h(t), t, \theta_f) dt. \tag{3}$$

Thus, a Neural ODE can transform from $h(0)$ to $h(T)$ through the solutions to the initial value problem (IVP) of the ODE. This framework indirectly realizes a functional relationship $x \to F(x)$ as a general neural network.

By the properties of ODEs, Neural ODEs are always invertible; we can reverse the limits of integration, or alternatively, integrate $-f$. The *Adjoint Sensitivity Method* (Pontryagin et al., 1961) based on reverse-time integration of an expanded ODE, allows for finding gradients of the initial value problem solutions $h(T)$ with respect to parameters $\theta_f$ and the initial values $h(0)$. This allows the training Neural ODE to use gradient descent, which allows them to combine with other neural network blocks.

## 4 LEARNING DYNAMICS ON SPHERICAL SPACE WITH NEURAL ODES

Consider a specific scenario where the dynamics unfold within a spherical space with a radius of $R = 1$, referred to as $\mathcal{S}$. In this context, it is known that the solution evolves within a submanifold of $\mathbb{R}^3$, and the vector field $f$ is defined on this submanifold. Let $\mathcal{G}$ represent a manifold learning function defined as follows: $\mathcal{G} : \mathbb{R}^3 \to \mathcal{S} \subset \mathbb{R}^3$. In simpler terms, $\mathcal{G}$ is a function that maps from three-dimensional Euclidean space to a submanifold $\mathcal{S}$ embedded within three-dimensional Euclidean space. Define $h$ as the state in three-dimensional Euclidean space, represented as $h = \begin{bmatrix} x \\ y \\ z \end{bmatrix} \in \mathbb{R}^3$. On the other hand, $l$ is the state within the submanifold, expressed as $l = \begin{bmatrix} u \\ v \end{bmatrix} \in \mathbb{R}^2$.

To establish a connection between the two representations, we can relate $u$ and $v$ to $h$ using the following equations: $h = \begin{bmatrix} x \\ y \\ z \end{bmatrix} = \begin{bmatrix} R \cdot sin(u)cos(v) \\ R \cdot sin(u)sin(v) \\ R \cdot cos(u) \end{bmatrix}$. The derivative of $h$ with respect to $t$ represents the rate of change of state $h$ with respect to time:

$$\frac{dh}{dt} = \frac{dh}{dl} \cdot \frac{dl}{dt} = \begin{bmatrix} \frac{\partial x}{\partial u} & \frac{\partial x}{\partial v} \\ \frac{\partial y}{\partial u} & \frac{\partial y}{\partial v} \\ \frac{\partial z}{\partial u} & \frac{\partial z}{\partial v} \end{bmatrix} \cdot \frac{dl}{dt} = R \cdot \begin{bmatrix} cos(u)cos(v) & -sin(u)sin(v) \\ cos(u)sin(v) & sin(u)cos(v) \\ -sin(u) & 0 \end{bmatrix} \cdot \begin{bmatrix} \frac{du}{dt} \\ \frac{dv}{dt} \end{bmatrix}. \quad (4)$$

Considering $\frac{dl}{dt}$ as the vector field within the manifold, we employ a neural network denoted as $f : l \rightarrow \frac{dl}{dt}$ to model this vector field. Function $f$ describes the evolution of the state $l$ within the manifold. Given an initial state $h(0)$ in the original space, we integrate $\frac{dh}{dt}$ over time to derive the final state $h(T)$:

$$h(T) = h(0) + \int_0^T \frac{dh}{dt}dt = h(0) + \int_0^T R \cdot \begin{bmatrix} cos(u)cos(v) & -sin(u)sin(v) \\ cos(u)sin(v) & sin(u)cos(v) \\ -sin(u) & 0 \end{bmatrix} \cdot f(l, t, \theta)dt. \quad (5)$$

The latent state $l$ within the manifold offers a more robust and expressive representation compared to the latent state $h$ within the original space. The evolution of vector fields is shown in Figure 2. The vector field learned by the Neural ODEs in manifold makes dynamic more explainable and easier to converge during learning. The trajectories of these dynamics are visualized in Figure 3. It also demonstrates the advantages of learning dynamics on manifold.

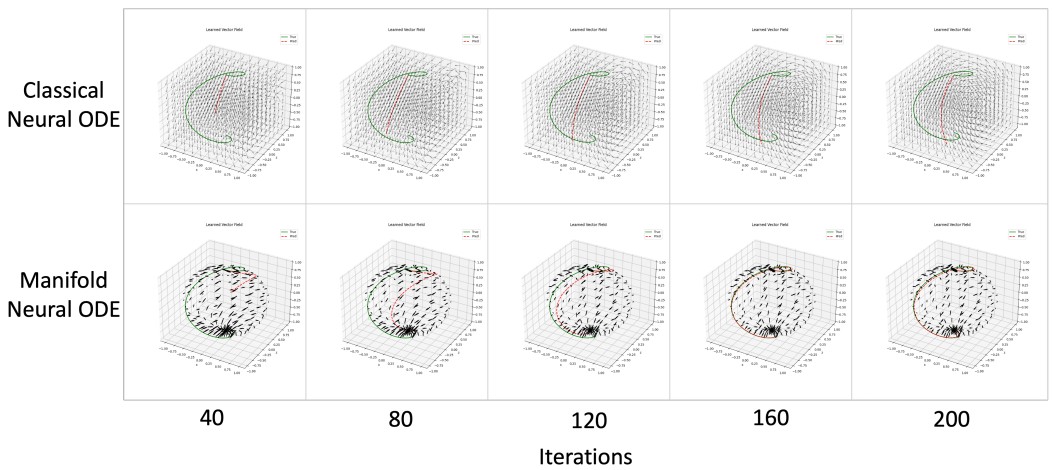

Figure 2: The vector fields are generated by Neural ODE in three-dimensional Euclidean space and by Manifold Neural ODE in spherical space. Each one is trained in 200 iterations.

## 5 LEARNING DYNAMICS ON MANIFOLDS WITH NEURAL ODES

Motivated by the previous case in Section 4, we now introduce a general methodology for learning dynamics on manifolds with Neural ODEs. As defined in Section 3.1, diffeomorphisms and diffeomorphic manifolds serve as fundamental constructs for our methodology.

**Theorem 1.** *Let $\mathcal{G} : \mathbb{R}^n \rightarrow \mathbb{R}^m$ be a manifold learning function, where $\mathcal{G}$ is assumed to be invertible, and $\mathcal{G}^{-1}$ denotes the inverse of function $\mathcal{G}$. Consider two sets of variables $h$ and $l$, representing latent states, with $h \in \mathbb{R}^n$ and $l \in \mathbb{R}^m$. Then, the derivative of $h$ with respect to time $\frac{dh}{dt}$ satisfies the equation: $\frac{dh}{dt} = J_{\mathcal{G}^{-1}} \cdot \frac{dl}{dt}$, where $J_{\mathcal{G}^{-1}}$ represents the Jacobian matrix of the inverse function $\mathcal{G}^{-1}$.*

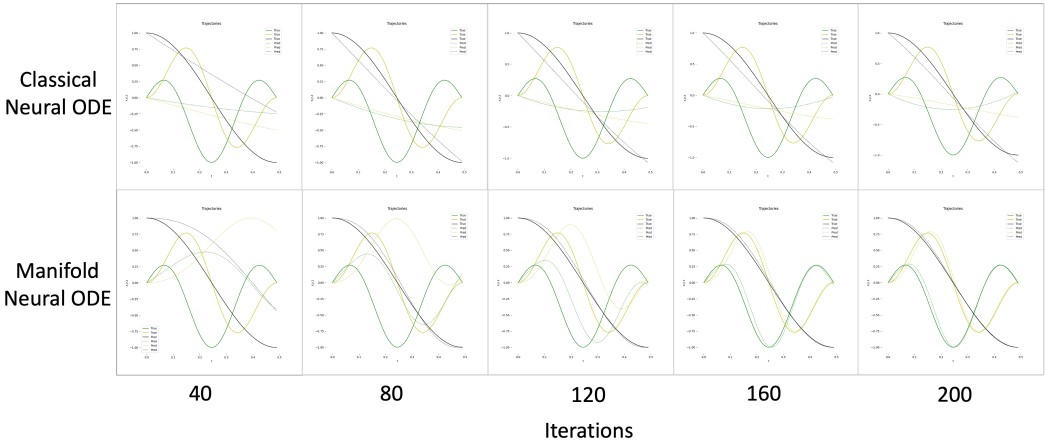

Figure 3: The trajectories learned by Neural ODE and Manifold Neural ODEs. The three different colors represent the three dimensions in the three-dimensional Euclidean space. The solid and dotted lines represent the true trajectories and learned trajectories, respectively.

*Proof.* We begin with the chain rule and obtain: $\frac{dh}{dt} = \frac{dh}{dl} \cdot \frac{dl}{dt}$. Using the property of the composition of functions and the inverse function, we have: $\frac{dh}{dl} = \frac{d(\mathcal{G}^{-1} \circ \mathcal{G}(h))}{dl} = \frac{d\mathcal{G}^{-1}(l)}{dl} = J_{\mathcal{G}^{-1}}$, where $J_{\mathcal{G}^{-1}}$ is the Jacobian matrix of function $\mathcal{G}^{-1}$. According to the inverse function theorem (Hamilton, 1982), the matrix inverse of the Jacobian matrix of an invertible function is the Jacobian matrix of the inverse function: $J_{\mathcal{G}^{-1}} = J_{\mathcal{G}}^{-1}$. Thus, we have: $\frac{dh}{dt} = J_{\mathcal{G}}^{-1} \cdot \frac{dl}{dt}$. $\qquad\square$

**Remark.** *This theorem provides a method for expressing the derivative of $h$ with respect to time $\frac{dh}{dt}$ in $\mathbb{R}^n$ in terms of the derivative of $l$ with respect to time $\frac{dl}{dt}$ in manifold $\mathbb{R}^m$, where $n > m$. It establishes a relationship between the dynamics of $h$ and $l$, enabling the transfer of information from the higher-dimensional space to the lower-dimensional manifold.*

Similar to classical Neural ODEs, we utilize a neural network $f$ to parameterize the vector field $\frac{dl}{dt}$. The key difference is that this vector field exists within the manifold. Then, we derive the representation of the vector field in the original space as follows:

$$\frac{dh}{dt} = J_{\mathcal{G}}^{-1} \cdot f(\mathcal{G}(h), t, \theta). \tag{6}$$

Given the initial state $h(0)$, we can easily obtain the solution of the dynamic at time $t = T$:

$$h(T) = h(0) + \int_0^T J_{\mathcal{G}}^{-1} \cdot f(\mathcal{G}(h), t, \theta) dt. \tag{7}$$

## 6 NUMERICAL EXPERIMENTS

We will demonstrate the superiority of our methodology in terms of accuracy, the number of function evaluations (NFEs), and convergence speed. In Section 6.1, we introduce the datasets and environment settings. In Section 6.2 we show on three real-life image datasets that our model has better prediction accuracy, fewer NFEs, and faster convergence speed compared to baselines. In Section 6.3, we show the results for three series datasets. All the models were implemented in Python 3.9 and realized in PyTorch. The experiments in Section 6.2 were conducted using a device equipped with an NVIDIA GeForce GTX 1070, while for the experiments in other sections, an Apple M2 with an 8-core CPU was used.

### 6.1 EXPERIMENTAL SETUP

**Datasets.** We evaluated our model with three image classification datasets and three series classification datasets. For the image classification task, we evaluate our model on the MNIST (Deng,

2012), CIFAR-10 (Krizhevsky et al., 2009), and SVHN (Netzer et al., 2011). MNIST is a hand-written digit database with a training set of $60,000$ samples. The CIFAR-10 training dataset consists of $60,000$ $32 \times 32$ color images in ten classes. SVHN is a digit classification dataset that contains $600,000$ $32 \times 32$ RGB images of printed digits (from 0 to 9) cropped from pictures of house number plates. For series datasets, we use BeetleFly, HandOutlines, and ECG200, which come from (Bagnall et al., 2018). BeetleFly is a dataset that distinguishes between beetles and flies, where the outline of the original image is mapped to a one-dimensional series at a distance from the center. HandOutlines is designed to test the efficacy of hand and bone outline detection and whether these outlines could be helpful in bone age prediction. ECG200 is a binary classification dataset that traces the electrical activity recorded during one heartbeat. The two classes are a normal heartbeat versus a myocardial infarction event.

**Evaluation metrics and baselines.** For the image classification task, we compared our model with NODEs and ANODEs in terms of training loss, test accuracy, NFEs, and convergence speed. We also compared ours with the ResNet with 10 residual blocks implemented in (Lin & Jegelka, 2018) in terms of test accuracy and training loss. For the series classification dataset, we compared our model with NODEs and ANODEs in terms of training loss, test accuracy, and NFEs. Despite the good performance, the main dedication of our approach is to provide a methodology that can benefit more models as a generalized approach, rather than just comparing it to baselines.

**Parameter settings.** For image datasets, we set the batch size as 32. We use the same vector field modeling in all continuous models. The vector field is modeled by three convolutional layers. The in channels and out channels for each layer are set as $(N_{in}, 32)$, $(32, 32)$, and $(32, N_{in})$ respectively, where the $N_{in}$ represents the number of channels of the input image. We use the $ReLU$ as the active function. We use the Adam algorithm as the optimizer with a learning rate of $10^{-3}$. To avoid experimental errors as much as possible, we run them three times for each model and record the corresponding mean and standard values. We run five epochs for each experiment since the experiment shows that five epochs are enough to converge. For Res-Net, we model it using 10 residual blocks, and each block is implemented by a two-layer MLP (Lin & Jegelka, 2018). For series classification tasks, we use the same vector field modeling in all continuous models. The vector field is implemented by a three-layer MLP with the hidden dimensions as 16. We run the experiment three times for each model and record the corresponding mean and standard values. We run 30 epochs for each experiment. For all the continuous models, we set the same tolerance of the ODE solver, as $10^{-3}$. For all the augmented models, we use five extra dimensions. For the manifold function $\mathcal{G}$, we implement it by two convolutional layers with the size of $(N_{in}, 16)$, $(16, 16)$, and a max pool layer which reduces the x-y dimensions by 2.

## 6.2 Image Classification

Considering images are usually in high dimensions, we apply our method to image classification tasks. MNIST dataset inherently inhabits a 784-dimensional space ($1 \times 28 \times 28$), and datasets like CIFAR-10 or SVHN, originally occupying a 3072-dimensional space ($3 \times 32 \times 32$). We employ an encoder to model the manifold function $\mathcal{G}$ and extract the Jacobian matrix from the corresponding decoder, denoted as $J_{\mathcal{G}}^{-1}$. Since the encoder and decoder can be automatically trained, we denote our model as AutoNODE. The AutoNODE learns the dynamics in a manifold space that is $16$ ($4 \times 4$) times less than the original space. We also test our methodology on the Augmented Neural ODE, denoted as Aug-AutoNODE.

**Test accuracy and training loss curves.** We test our methodology on Neural ODEs and Augmented Neural ODEs. We also compare them with ResNet. The test accuracy is shown in Table 1. The results show that our model can achieve the lowest training loss, highest test accuracy in the three datasets. Besides, our model can continue to converge at a stable and fast rate.

**Number of function evaluations.** Since our methods aim to learn a simpler vector field within a manifold, they require fewer function evaluations than NODEs and ANODEs. To test this, we measure the NFEs in both the forward evaluation and backpropagation processes. We visualize the NFEs in Figure 4. Our models' NFEs can maintain a relatively stable level whereas the baseline models' NFEs will increase rapidly with the training iteration increase. This is also one of the reasons that the baseline model has a slow convergence rate. We show the average NFEs over three runs in Table 1.

Table 1: Testing Accuracy and Average NFEs in Image Classification

|  | Testing Accuracy | | | Average NFEs | | |
|---|---|---|---|---|---|---|
|  | CIFAR-10 | MNIST | SVHN | CIFAR-10 | MNIST | SVHN |
| ResNet-10 | $0.437 \pm 0.007$ | $0.978 \pm 0.001$ | $0.604 \pm 0.017$ | − | − | − |
| NODE | $0.512 \pm 0.011$ | $0.944 \pm 0.005$ | $0.758 \pm 0.036$ | 61.07 | 61.87 | 82.50 |
| ANODE | $0.568 \pm 0.008$ | $0.981 \pm 0.001$ | $0.568 \pm 0.008$ | 70.88 | 43.76 | 93.03 |
| AutoNODE (Manifold) | $0.663 \pm 0.012$ | $0.988 \pm 0.002$ | $0.872 \pm 0.003$ | 39.89 | 40.08 | 47.12 |
| Aug-AutoNODE (Manifold) | $\mathbf{0.672} \pm 0.003$ | $\mathbf{0.989} \pm 0.001$ | $\mathbf{0.874} \pm 0.004$ | **39.16** | **39.92** | **40.81** |

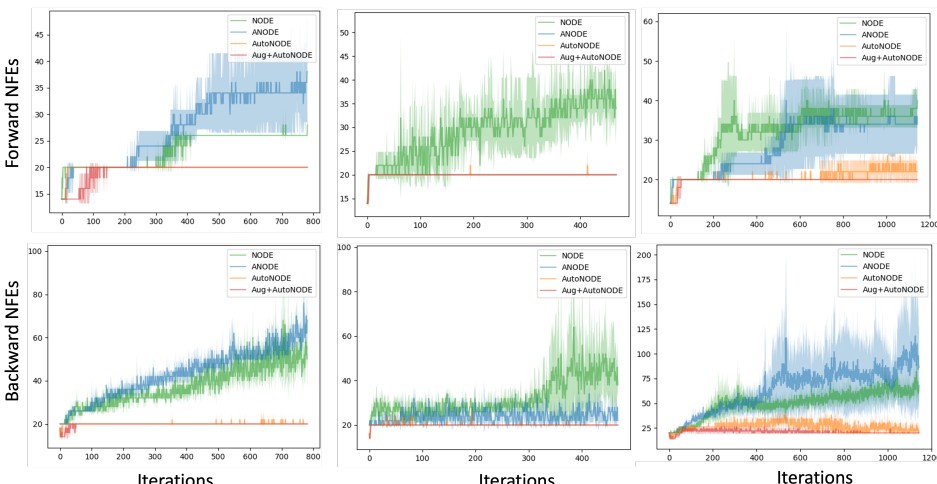

Figure 4: The top three plots show the number of function evaluations (NFEs) in the forward evaluation and the bottom three plots show the NFEs in backpropagation processes on CIFAR-10, MNIST, and SVHN.

**Time and convergence speed.** To better compare the time and convergence speed, we record the absolute time for each iteration. We plot the corresponding training loss and test accuracy over time during the training process. Figure 5 shows the training iteration loss over time. Figure 6 shows the training epoch loss and test accuracy over time. Our models typically require only $\frac{1}{3}$ to $\frac{1}{2}$ of the time of the baseline model. Thus our models have faster convergence speed and require little training time as well as achieve better test accuracy.

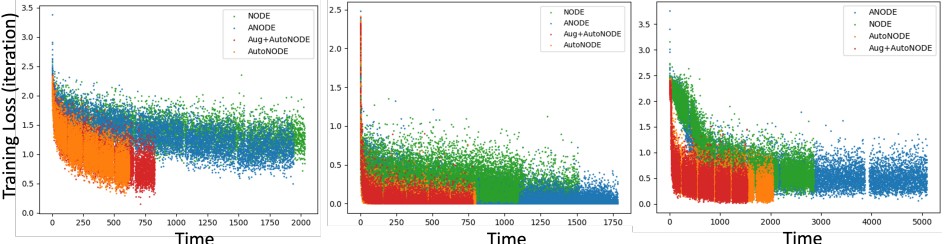

Figure 5: The corresponding training iteration loss for each model over time during the training. From left to right, CIFAR-10, MNIST, and SVHN are used, respectively.

## 6.3 SERIES CLASSIFICATION

For series classification tasks, the test accuracy and average NFEs are recorded in Table 2. Since the dataset is not as complex as the image dataset, the NFEs for all the models are relatively few. The Augmented AutoNODE has maintained 14 NFEs at all times and thus it needs the least computation for the dynamics. Our models have the best test accuracy with a more stable and faster training convergence speed.

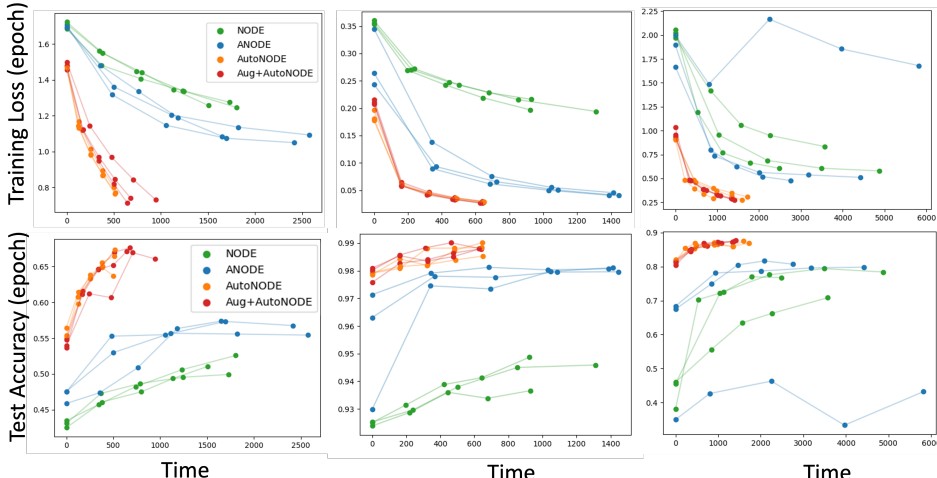

Figure 6: The corresponding training epoch loss and the test accuracy for each model over time during the training. From left to right, CIFAR-10, MNIST, and SVHN are used, respectively. Each experiment runs three times.

Table 2: Testing Accuracy and Average NFEs in Series Classification

|  | Testing Accuracy | | | Average NFEs | | |
| --- | --- | --- | --- | --- | --- | --- |
|  | BeetleFly | HandOut | ECG200 | BeetleFly | HandOut | ECG200 |
| NODE | $0.800 \pm 1.110$ | $0.897 \pm 0.007$ | $0.836 \pm 0.011$ | **14.00** | 16.87 | 14.15 |
| ANODE | $0.816 \pm 0.023$ | $0.888 \pm 0.011$ | $0.836 \pm 0.006$ | **14.00** | 14.44 | 14.00 |
| AutoNODE (Manifold) | **0.867** $\pm 0.023$ | **0.915** $\pm 0.003$ | **0.862** $\pm 0.004$ | **14.00** | 16.28 | 14.90 |
| Aug-AutoNODE (Manifold) | **0.867** $\pm 0.023$ | $0.912 \pm 0.003$ | $0.849 \pm 0.010$ | **14.00** | **14.00** | **14.00** |

## 7 CONCLUSION

In this work, we introduced a novel approach to address the challenges of learning dynamics in high-dimensional systems. By integrating manifold learning principles with Neural ODEs, our method offers an efficient and accurate solution for the dynamic learning. We leveraged the manifold hypothesis and project the original data into the manifold by an encoder, and leverage the Jacobian matrix of its corresponding decoder for recovery. Our methodology allows us to reduce complexity while preserving accuracy in dynamic learning. Experimental evaluations across diverse datasets consistently demonstrated our approach's superiority, underscoring its potential to advance our understanding of high-dimensional dynamic systems and improve modeling accuracy.

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
