# OpenReview forum: "Learning Dynamics on Manifolds with Neural Ordinary Differential Equations"
_ICLR.cc/2024/Conference — ICLR 2024 Conference Withdrawn Submission_

### Official Review · Reviewer_ivZK · 2023-10-27

**Soundness:** 1 poor
**Presentation:** 4 excellent
**Contribution:** 1 poor
**Rating:** 3
**Confidence:** 4

**Summary:**

The paper proposes a model to learn jointly a manifold and the dynamics on this manifold. The manifold is learned through an auto-encoder. Dynamics are learned by a neural ODE. The model is evaluated on classification tasks on several image and time series datasets.

**Strengths:**

The paper is clearly written and easy to follow. It tackles the important question of reducing the computational cost of neural ODEs by learning neural ODEs on lower-dimensional manifolds. The experiments suggest that the proposed model requires less function evaluations and obtains better classification performance than the standard neural ODE approach.

**Weaknesses:**

I have two important concerns with the paper: novelty with respect to the literature, and soundness of the proposed model.

**Novelty:**

The idea of combining neural ODEs with manifold learning is already present in several papers (some of which are cited by the present paper in Section 2). The paper should make a more precise comparison with the literature, beyond stating that the literature focuses on the problem of sampling and not on the one of classification. To be more precise:
1. I do not understand how Theorem 1 from the present paper differs from Proposition 5.1 of [Lou et al., 2020].
2. The idea to use auto-encoders in conjunction with neural ODEs is already present in the community of machine learning for physics, see e.g.

Deep learning models for global coordinate transformations that linearise PDEs, Gin, C., Lusch, B., Brunton, S., & Kutz, J. (2021).

Physics-informed neural ODE (PINODE): embedding physics into models using collocation points, Sholokhov, A., Liu, Y., Mansour, H. et Nabi S. (2023).

As a consequence, the main contribution of the paper in my opinion is to show empirically on several classification datasets that the proposed method does indeed reduce the number of function evaluations while improving the performance. This should be reflected in the end of the introduction (and probably also in the title of the paper), which should be more modestly phrased in my opinion.

**Soundness:**

Solving an ODE on a manifold is a difficult task (see [Hairer, 2011]). In particular, even though the solution of the ODE defined by eq. (7) belongs to the manifold defined by $G$, its discretization through a solver may not belong to the manifold.

This issue is not mentioned in the paper, and is further exacerbated by the fact that the paper does not assume exact access to the inverse Jacobian of $G$, but reconstructs it using the decoder.

But the paper does not mention a term in the loss that would ensure that the encoder-decoder approximates the identity mapping. Assuming that such a term was not introduced in the loss, it seems very unlikely that the decoder can be used to approximate the inverse Jacobian of $G$. In this case, I do not see the difference between the proposed model and a standard neural ODE parametrized by a convolutional network with downsampling and upsampling. If an additional term to force the encoder-decoder to learn the identity mapping was introduced, this should absolutely be mentioned, and the choice of associated hyperparameters should be discussed.

For all these reasons, I think that the paper is unfit for acceptance at this stage.

**Minor comments that did not influence the rating:**
1. Plots are hard to read for colorblind people / when printed in black&white. It would be nice when possible to make them more colorblind-friendly.
2. The introduction of neural ODEs in Section 3.2 is somewhat imprecise. The link between residual networks and neural ODEs is much more complicated than presented by the authors (see references below for more precise mathematical statements regarding this link). I do not think that it is useful to refer to residual networks to introduce neural ODEs, since they are an interesting standalone model independently of residual neural networks and since the connection is actually quite subtil.

Scaling Properties of Deep Residual Networks, Cohen A.S., Cont R., Rossier A., Xu R. (2021).

Scaling ResNets in the Large-depth Regime, Marion P., Fermanian A., Biau G., Vert J.P. (2022).

**Questions:**

I have three questions for the authors:

1. Have you used a term in the training loss to ensure that the encoder-decoder approximates the identity mapping? If so, did you perform experiments that show it is indeed the case that the identity mapping is well approximated by the auto-encoder after training?

2. As far as I understand, eq. (7) is equivalent to

$x(0) = G(h(0))$

$x(T) = x(0) + \int_0^T f(x(t), t, \theta) dt $

$h(T) = G^{-1}(x(T))$

Can you confirm, and explain the difference if there is one?
Furthermore, if the two formulations are indeed equivalent in the continuous world, it may not be the case anymore once they are discretized. Have you tried training with the alternative formulation? It seems more natural since the ODE solver is then called on the low-dimensional object instead of the high-dimensional one.

3. How do you deal with time-series data in your experiments? In its simplest form, neural ODEs are suited for vector data, which is inputed in the first layer. Of course, many proposed modifications in the literature make it suitable for time series, but I do not think this is mentioned in the paper, and was wondering how you are handling this.

---

### Official Review · Reviewer_V8a6 · 2023-11-01

**Soundness:** 2 fair
**Presentation:** 1 poor
**Contribution:** 2 fair
**Rating:** 1
**Confidence:** 4

**Summary:**

This manuscript proposes a method to further improve ODE-inspired NN models such as Neural ODE by casting the learning to a manifold. The hypothesis is that by doing so, the learning can be more efficient, for example, as measured by the number of function evaluations (NFE). The work is motivated by Floryan and Graham, Nature Machine Intelligence, 2022

**Strengths:**

The paper follows the same direction as Floryan & Graham 2022, which is an emerging research direction when modeling complex system dynamics. In stead of AE, the paper proposes to use NODE as the learning vehicle. The idea is novel. The authors provided some empirical evidence on the effectiveness of the method.

**Weaknesses:**

The manuscript can be vastly improved. Below is a list of general issues. The detailed questions are listed in the next section.

1. The authors seem to have a different understanding on the meaning of `learning dynamics from data`. As in Floryan and Graham, the concept is to learn the dynamic behavior from sampled data, and to generalize to other cases. However, the experimental results are all point out to image and time series classification. What type of dynamics are the models supposed to learn?

2. Sections 4 and 5 are the key components of the paper. However as it is written, many details are missing. The manuscript provides an parameterized version of $\mathbb{R}^3 \rightarrow \mathbb{R}^2$.  It is unclear how $\mathcal{G}$ in Section 5 is defined.

3. In the experimental results section, the authors discussed the dimensions of the input space, e.g., $748$ for MNIST. But they are not dynamic systems, what does the learning of dynamics mean for these cases?

4. What is the connection between the accuracy of a trained model versus the training on a manifold, as shown in Table 1?

**Questions:**

1. Figure 3 is difficult to read. The captions are also not clear. What exactly is shown in the figures?

---

### Official Review · Reviewer_ckcW · 2023-11-01

**Soundness:** 2 fair
**Presentation:** 2 fair
**Contribution:** 2 fair
**Rating:** 5
**Confidence:** 4

**Summary:**

The authors introduce a data-driven approach that employs an encoder to project the original data into a manifold. They use the Jacobian matrix of a corresponding decoder for mapping back to the original data space. Based on this, the authors extend the concept of Neural ODEs to operate on manifolds.

**Strengths:**

The proposed method of mapping data into a submanifold and using Jacobian to map it back is clear and straightforward.

**Weaknesses:**

1. In Section 4, please specify the submanifold of which is referred to. As the authors suggest in Section 4, $u$ and $v$ stand for the $\theta$ and $\phi$ of the spherical coordinate.
    - If authors are referring to the submanifold of $S^2$, all pairs of $(u,v)$ do not form a submanifold of $S^2$ as at the north and south poles, multiple pairs $(u,v)$ map to the same point, causing the Jacobian to lose rank.
    - If authors are referring to the submanifold of $\mathbb{R}^3$, all pairs of $(u,v)$ do not reside in $\mathbb{R}^3$. A submanifold usually denotes a subset of the manifold that itself has a manifold structure. All pairs of $(u,v)$ together with $R$ is used to parameterize $(x,y,z)$ but all pairs of $(u,v)$ themselves are not subset of $\mathbb{R}^3$.
2. In Section 5, the authors claim a general methodology for learning dynamics on manifolds. However, no experiments of data on manifolds with specific submanifolds are conducted to justify the effectiveness. The authors should at least provide experiments on the dynamics of sphere data to align with Section 4.

**Questions:**

See weaknesses.

---

### Official Review · Reviewer_Ha6r · 2023-11-01

**Soundness:** 3 good
**Presentation:** 3 good
**Contribution:** 3 good
**Rating:** 3
**Confidence:** 3

**Summary:**

- The paper introduces a novel approach that combines manifold learning principles with Neural Ordinary Differential Equations (ODEs) to address the challenges of learning dynamics in high-dimensional systems.
- The proposed method leverages the manifold hypothesis and projects the original data into the manifold using an encoder. The Jacobian matrix of the corresponding decoder is used for dynamics learning.
- Experimentally, the results show that the proposed method achieves better test accuracy, faster convergence speed, and requires fewer function evaluations (NFEs) compared to the baseline models.

**Strengths:**

- Theoretical in the simple low dimensional space, the proposed method is sound and align with the Neural ODE. The paper introduces a novel approach that combines manifold learning principles with Neural ODEs to address the challenges of learning dynamics in high-dimensional systems. This integration allows for efficient and accurate solutions for dynamic learning.
- The results show that the models based on the proposed approach have faster convergence speed, require less training time, and achieve better test accuracy compared to baseline models.

**Weaknesses:**

- Experiments are very simple. The experiments are based on either small dataset or simple image dataset (MNIST/ Cifar-10/ etc).
- Comparison is not so convincing. For example, normally for the image dataset, we use resnet-18 with convolutional layers instead of simple fully-connected layers. This makes the performance of resnet-10 bad. However, this also limits the contribution of the proposed method, as this only shows small improvement over this simple baseline.

**Questions:**

- What is the main reason for not using normal resnet-18/ other CNN as comparisons?